# Identifying the Mutual Correlations and Evaluating the Weights of Factors and Consequences of Mobile Application Insecurity

Elena Zaitseva [1], Tetiana Hovorushchenko [2,*], Olga Pavlova [2] and Yurii Voichur [2]

1    Department of Informatics, University of Zilina, 01026 Zilina, Slovakia; elena.zaitseva@fri.uniza.sk
2    Department of Computer Engineering and Information Systems, Khmelnytskyi National University, 29016 Khmelnytskyi, Ukraine; pavlovao@khmnu.edu.ua (O.P.); voichury@khmnu.edu.ua (Y.V.)
*    Correspondence: hovorushchenko@khmnu.edu.ua

**Abstract:** Currently, there is a contradiction between the growing number of mobile applications in use and the responsibility that is placed on them, on the one hand, and the imperfection of the methods and tools for ensuring the security of mobile applications, on the other hand. Therefore, ensuring the security of mobile applications by developing effective methods and tools is a challenging task today. This study aims to evaluate the mutual correlations and weights of factors and consequences of mobile application insecurity. We have developed a method of evaluating the weights of factors of mobile application insecurity, which, taking into account the mutual correlations of mobile application insecurity consequences from these factors, determines the weights of the factors and allows us to conclude which factors are necessary to identify and accurately determine (evaluate) to ensure an appropriate level of reliability of forecasting and assess the security of mobile applications. The experimental results of our research are the evaluation of the weights of ten OWASP mobile application insecurity factors the identification of the mutual correlations of the consequences of mobile applications' insecurity from these factors, and the identification of common factors on which more than one consequence depends.

**Keywords:** mobile application security; mobile application insecurity; mobile application insecurity factors; mobile application insecurity consequences; mutual correlations of consequences from insecurity factors; weights of insecurity factors

## 1. Introduction

Currently, we cannot imagine our daily life without a smartphone and the numerous applications that we use for various purposes. Nowadays, people increasingly rely on mobile applications for all aspects of their lives and use them several times a day. The Apple App Store and Google Play Store offer more than eight million different applications. However, we cannot be sure that the program came from a reputable source and that it is completely safe. Despite Apple's procedure before apps are released in the App Store and Google's procedure before apps are released in Google Play, there are still many examples of mobile app vulnerabilities and security risks, and in particular, the data processed by apps and mobile devices are usually targeted by cybercriminals. Mobile operating systems lack effective tools for detecting malware that compromises personal data [1]. Mobile applications can potentially pose serious security risks to their users. Applications may contain vulnerabilities that can be exploited by attackers to gain unauthorized access to device resources, including confidential information on a mobile device [2].

Thus, mobile applications are a critical component of the mobile ecosystem that requires further research to develop the security methods and tools necessary to mitigate the risks associated with mobile applications.

The vulnerabilities of mobile applications (authentication and authorization errors, data leakage, etc.) and their security risks (API vulnerabilities, weak authorization and

authentication, client-side injection, a low level of server-side security, insecure data storage, insecure data transmission, data leakage, improper session handling, the use of a broken or insecure encryption algorithm, etc.) are very dangerous because today, users are used to trusting confidential information to their devices, including financial information and medical data, which is a serious cybersecurity challenge for developers and suppliers of mobile applications [3,4], as the data processed by mobile applications and devices are usually targeted by cybercriminals [5]. In addition, as many new mobile applications on the Internet of things (IoT) have emerged recently, the threat posed by wormhole attacks has increased [6]. In general, security is one of the main challenges in mobile IoT networks, which consist of a huge number of objects, leading to an increase in the number of threats in our daily lives [7,8]. Today's society is so dependent on networks of mobile Internet of things that cyberphysical attacks are considered a key threat to ordinary citizens, endangering both material values and human lives [9,10].

According to statistics [11], 312 cases of Android application vulnerabilities and 87 cases of iOS application vulnerabilities were recorded. In general, the open-source nature of Android makes this operating system and Android applications a prime target for malware [12–14]. The Android intercomponent communication (ICC) mechanism can cause security issues such as application security policy violations, especially in the case of interapp communication (IAC) [3]. Currently, there is still a need for methods to select attributes for use in Android malware detection systems [15,16].

According to NowSecure's benchmark testing [17], 85% of the applications studied had one or more security risks. More than 50% of the investigated applications had bottlenecks that led to data protection problems during transmission. About one-third of the tested applications had problems with the source code. In particular, Android apps had code problems that could lead to reverse engineering and other threats.

According to [17,18], when it comes to mobile App security, the main problems that occur the most frequently are improper platform usage, insecure data storage, insecure client–server communication, insecure authentication (for example, the password authentication of users imposes a number of restrictions and is no longer considered safe and convenient for mobile users, while biometric authentication of users has recently attracted increasing attention as a promising solution for improving mobile security [19,20]), insecure authorization, insufficient data encryption, poor code quality, code tampering, reverse engineering risk, and extraneous functionality. The frequency of these precedents impact on the security of mobile applications is shown in Figure 1 (on the basis of the OWASP Mobile Top 10 Risks from 2018 [17,18], because the new Mobile Top 10 Risks list for 2023 is being worked upon, as indicated on the official OWASP website). Next, in this paper, only these ten most prominent factors (OWASP Mobile Top 10 Risks) that affect mobile application security are analyzed.

Figure 1 shows that insecure data storage and insecure client–server communication are the factors that most affect the security of mobile applications. However, the influence of other factors should not be underestimated too.

Thus, the development of modern mobile application development technologies requires the dynamic development of methods and tools to ensure the security of mobile applications. For example, a timely forecasting of mobile application security can be used to take any preventive measures to reduce the number of application vulnerabilities, security risks, etc. Currently, there is a contradiction between the growing number of mobile applications in use and the responsibility that is placed on them, on the one hand, and the imperfection of the methods and tools for ensuring the security of mobile applications, on the other hand. Therefore, ensuring the security of mobile applications by developing effective methods and tools is a challenging task today. Our study is actually devoted to evaluating the weights of factors and identifying the mutual correlations of mobile application insecurity consequences, when developing effective methods and tools for assessing and forecasting the security of mobile applications.

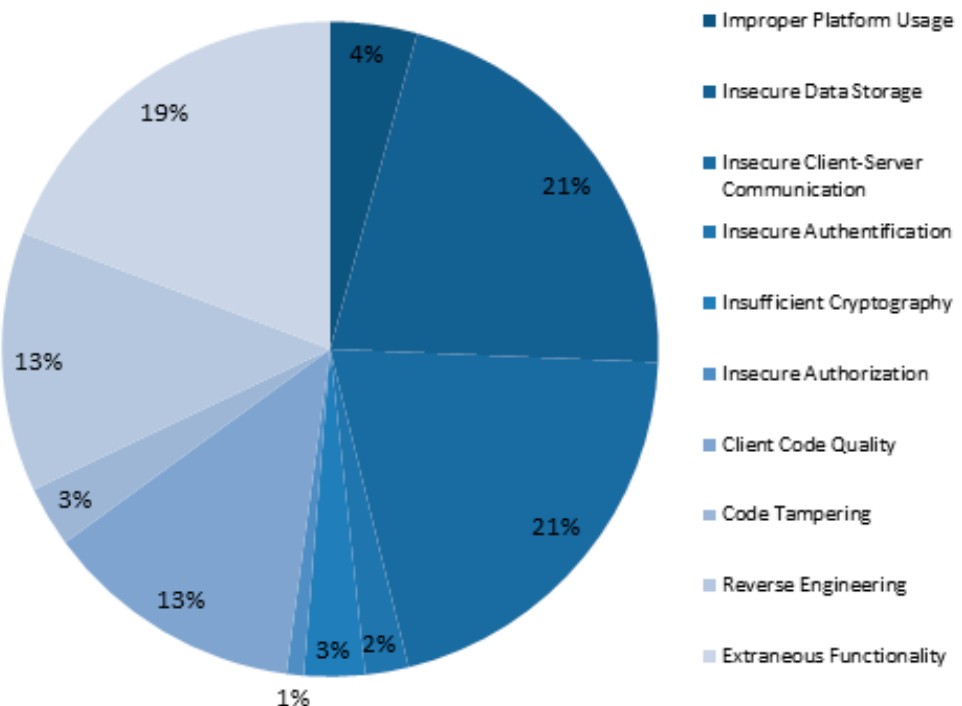

**Figure 1.** The Frequency of Threat Factors Influences on the Security of Mobile Applications [17,18].

The remaining portion of the paper is structured as follows: The Section 2 provides a comprehensive state of the art of the related works being carried out. The Section 3 details the proposed methodology. The Section 4 details the implementation of the proposed method and a discussion about the application of the obtained results. The final section concludes the paper and describes future enhancements.

## 2. State of the Art

In the course of the study, an analysis of the influence of the factors indicated in Figure 1 was carried out on examples from real cases of mobile applications used by millions of users, and the consequences that the influence of the above-mentioned factors led to. The results of the analysis are shown in Table 1.

**Table 1.** Impact of Threat Factors on Mobile Application Security on Examples from Real Cases.

| Company or Application Name | Cause or Factor of the Threat | Consequences |
|---|---|---|
| Tinder, OkCupid, Bumble dating applications [21] | Insecure data storage | Popular dating apps such as Tinder, OkCupid, and Bumble have vulnerabilities that make users' personal information potentially available to stalkers, spammers, and hackers. Security breaches, which vary in severity and scope, can expose people's names, logins, locations, message history, and other account activity. |
| Fitness Balance app, Heart Monitor, Calories Tracker app [22] | Improper platform usage bypasses apple's iOS Touch ID security system | Once you scan your fingerprint, the apps briefly display an in-app purchase pop-up, charging $90 to $120, while dimming the screen to make it hard to see the tip. In some cases, even if you refuse to use Touch ID to enable the feature, the app asks you to tap to continue and instead attempts an in-app payment scam. |
| Children's smart watches with GPS (R7-2019-57) [23] | Insecure client–server communication: interception of sensitive data in transit over the network | The watches were supposed to be contacted using approved contact numbers via a whitelist mode, but the company discovered that the filters did not even work. The watch even accepted customization commands via text messages. This meant a hacker could change the watch's settings and put children at risk. |

**Table 1.** *Cont.*

| Company or Application Name | Cause or Factor of the Threat | Consequences |
| --- | --- | --- |
| Hacking of a US bank in 2019 due to a flaw in the bank's website and bypassing two-factor authentication by a cyberattacker [18] | Insecure authentication risk | The attacker logged in with the victim's stolen credentials, and when taken to a page that required a PIN or security response, the attacker used a modified string in the web address that set the computer up as recognized. This allowed them to cross the stage and start electronic transfers. |
| Philips HealthSuite Health Android app [24] | Insufficient data encryption | The issue, which was traced to insufficient encryption reliability, opened the app up to hackers who could access users' heart rate, blood pressure, sleep status, weight and body composition, and more. |
| Pandora, a smart car alarm system [25] | Insecure authorization risk | Stealing a smart alarm user account is not only possible but not that difficult. You do not even need to buy the alarm itself (which can cost a hefty $5000) to steal a Viper or Pandora account. At the time of the study, all you had to do to access the system was to register an account on the website or app and use it to access any other account. |
| WhatsApp Messenger [26] | Poor code quality | Until recently, WhatsApp had a serious vulnerability that was exploited by attackers to remotely install malware that would monitor "selected" smartphones after making a WhatsApp audio call to them. A WhatsApp exploit that installed Pegasus spyware on Android and iOS devices was discovered and adopted by the Israeli company NSO Group (the maker of the most advanced software tool). |
| Target app from any application store [18,27] | Code forgery risk | The attacker uses code modification through malicious forms of mobile applications, available in app stores, which may resort to phishing attacks to force the user to install the application. |
| Pokemon Go mobile game [28] | Reverse engineering | An attacker typically downloads a target app from an app store and analyzes it in their local environment using a variety of tools. After that, they can change the code and change the function of the app. Pokemon Go suffered a security breach when it emerged that users had re-engineered the app to know when Pokemon were nearby and catch them within minutes. |
| The idea behind the Wi-Fi File Transfer application was to open a port on Android and allow a connection from a computer [29] | Extraneous functionality | A group of researchers from the University of Michigan discovered hundreds of apps in Google Play that performed an unexpected trick: by effectively turning the phone into a server, they allowed the owner to connect to that phone directly from their computer, just like a website or other Internet service. However, dozens of these apps left unprotected ports open on these smartphones. This allowed attackers to steal data, including contacts or photos, or even install malware. |

In addition to the mobile application security cases listed in Table 1, a number of other studies have been conducted on the issue of apps' insecurity.

For example, Encalada et al. [30] analyzed the problem of mobile apps' information security and privacy issues from a theoretical and empirical perspective. The theoretical analysis showed a poor knowledge of users about the security of the shared information and a growing concern about the given misuse. The empirical results revealed high levels of insecurity in users about mobile applications being given access to their information.

Phasinam and Kassanuk [31] established a taxonomy based on the application domain and the architectural design to better detect IoT security problems. Using network host scanning and vulnerability scanning technologies, raw data on IoT apps were obtained. However, this paper only provided an in-depth study of apps' attacks and vulnerabilities related to the agriculture field.

Amelang [32] raised the issues of privacy and data security of a mobile application, which presented a digitized, "smart" version of a menstruation calendar. In particular, this study addressed two forms of insecurity—the unanticipated collection of user data by

private companies and the potential surveillance. However, these forms of app insecurity mainly related to the gynecology field, although it reflected a general trend of concern about data security in mobile applications.

Chakraborty et al. [33] aimed to find determinants of data-privacy anxiety among Indians and to understand their anxiety during the use of digital applications in their daily routines. Emerging themes from the data indicated the systemic determinants of data-security anxiety.

Aljumah, Altuwijri et al. [34] noted that determining the security of mobile applications was a difficult task and especially paid attention to trying to spot insecure apps, by analyzing user feedback on the Google Play platform and using sentiment analysis to determine the apps' level of security.

The research in [35] tried to delve deep into the innards of Android to come up with some plausible solutions to somewhat assuage the damage that three intentional leaks of the operating system could create. The paper proposed the ASBP framework (App Sanitization and Better Permissions), which would help remove the identified vulnerabilities that afflict Android, namely third-party apps and vendor-customized ones.

Sanni, Akinyemo et al. [36] is devoted to the development of a predictive model to detect and prevent suspicious customers with cyberthreat potential during the onboarding process for Mobile Money Services in developing nations using a machine learning (ML) technique with the purpose of minimizing the risks of cybercrime.

The goal of Weichbroth et al. [37] was the identification and analysis of the existing threats and best practices in the domain of mobile security. The study results contributed to the theory on mobile security through the identification and exploration of a variety of issues, regarding both threats and best practices.

Therefore, it can be concluded from Table 1 and other considered research that it is relevant and important to study the mutual correlation of risk factors and the consequences they entail for the development of methods and tools for ensuring and predicting the security of mobile applications. Thus, the purpose of our research was the evaluation of the mutual correlations and weights of the factors and consequences of mobile application insecurity.

### 3. Proposed Methodology

*3.1. Modelling the Subject Area of Mobile Application Security Assessment and Forecasting*

According to Figure 1, there are ten most prominent factors that affect mobile application security.

We analyzed the cases and examples presented in Table 1 and detected that the factors shown in Figure 1 led to consequences that are summarized in the following.

The consequences of mobile application security insufficiency are:

1. Excessive memory usage;
2. Reputation damage;
3. Material loss;
4. Program unpredictable crashes;
5. Fraud;
6. Identity theft;
7. Information leaks;
8. Slow loading of UX graphic elements;
9. Privacy violation;
10. Code theft;
11. Unauthorized access to data;
12. Intellectual property theft;
13. Information theft;
14. External policy violation;
15. Error occurrence.

The development of effective methods and tools for ensuring the security of mobile applications requires effective methods and tools for assessing and forecasting the security of mobile applications. To effectively assess and forecast the security of mobile applications, it is necessary to forecast the consequences—for example, a probabilistic assessment of their occurrence based on assessments of existing factors. The existence of relationships (correlations) between factors and consequences of mobile application insecurity affects the significance and weight of these factors. It is possible to emphasize the greater use and therefore the greater significance of the correctness of the assessment of some factor by taking into account their weights when assessing and forecasting the security of mobile applications.

According to information from [18,21–29], let us represent the dependencies of the mobile application insecurity consequences on the factors. Formulas (1)–(15) give the dependencies of the mobile application insecurity consequences "reputation damage" (rd), "material loss" (ml), "identity theft" (it), "fraud" (fr), "information theft" (itt), "error occurrence" (eo), "excessive memory usage" (emu), "information leaks" (il), "privacy violation" (pv), "code theft" (ct), "intellectual property theft" (ipt), "external policy violation" (epv), "unauthorized access to data" (uad), "program unpredictable crashes" (puc), and "slow loading of UX graphic elements" (slux) from the factors "improper platform usage" (ipu), "insecure data storage" (ids), "insecure client–server communication" (icsc), "insecure authentication risk" (iar), "insecure authorization risk" (iazr), "extraneous functionality risk" (efr), "code forgery risk" (cfr), "poor code quality" (pcq), "insufficient data encryption" (ide), and "reverse engineering risk" (rer):

$$rd = f_1(ids, icsc, iar, iazr, cfr, pcq, ide, rer) =$$
$$= \varphi_1(w_2 \cdot ids, w_3 \cdot icsc, w_4 \cdot iar, w_5 \cdot iazr, w_7 \cdot cfr, w_8 \cdot pcq, w_9 \cdot ide, w_{10} \cdot rer), \tag{1}$$

where $w_i$ ($i = 1 \ldots 10$) indicate the weights of 10 mobile application insecurity factors, and $f_j$, $\varphi_j$ ($j = 1 \ldots 15$) represent the functions of dependencies;

$$ml = f_2(ipu, ids, cfr, rer) = \varphi_2(w_1 \cdot ipu, w_2 \cdot ids, w_7 \cdot cfr, w_{10} \cdot rer), \tag{2}$$

$$it = f_3(ipu, ids) = \varphi_3(w_1 \cdot ipu, w_2 \cdot ids), \tag{3}$$

$$fr = f_4(ipu, ids, iazr) = \varphi_4(w_1 \cdot ipu, w_2 \cdot ids, w_5 \cdot iazr), \tag{4}$$

$$itt = f_5(icsc, iar, iazr, cfr, ide) = \varphi_5(w_3 \cdot icsc, w_4 \cdot iar, w_5 \cdot iazr, w_7 \cdot cfr, w_9 \cdot ide), \tag{5}$$

$$eo = f_6(cfr, pcq) = \varphi_6(w_7 \cdot cfr, w_8 \cdot pcq), \tag{6}$$

$$emu = f_7(pcq) = \varphi_7(w_8 \cdot pcq), \tag{7}$$

$$il = f_8(ide, rer) = \varphi_8(w_9 \cdot ide, w_{10} \cdot rer), \tag{8}$$

$$pv = f_9(ipu, efr, ide) = \varphi_9(w_1 \cdot ipu, w_6 \cdot efr, w_9 \cdot ide), \tag{9}$$

$$ct = f_{10}(cfr, ide) = \varphi_{10}(w_7 \cdot cfr, w_9 \cdot ide), \tag{10}$$

$$ipt = f_{11}(efr, ide, rer) = \varphi_{11}(w_6 \cdot efr, w_9 \cdot ide, w_{10} \cdot rer), \tag{11}$$

$$epv = f_{12}(ids, efr, rer) = \varphi_{12}(w_2 \cdot ids, w_6 \cdot efr, w_{10} \cdot rer), \tag{12}$$

$$uad = f_{13}(icsc, efr, cfr, rer) = \varphi_{13}(w_3 \cdot icsc, w_6 \cdot efr, w_7 \cdot cfr, w_{10} \cdot rer), \tag{13}$$

$$puc = f_{14}(iar, cfr, pcq) = \varphi_{14}(w_4 \cdot iar, w_7 \cdot cfr, w_8 \cdot pcq), \tag{14}$$

$$slux = f_{15}(pcq) = \varphi_{15}(w_8 \cdot pcq). \tag{15}$$

Therefore, the set of mobile application insecurity factors (MAIF) has the form:

$$MAIF = \{maif_1, \ldots, maif_{10}\} = \{ipu, ids, icsc, iar, iazr, efr, cfr, pcq, ide, rer\}. \tag{16}$$

The set of weighted factors of mobile application insecurity (MAIFW) has the form:

$$MAIFW = \{maifw_1, \ldots, maifw_{10}\} = \{ w_1 \cdot ipu, w_2 \cdot ids, w_3 \cdot icsc, w_4 \cdot iar, w_5 \cdot iazr, \\ w_6 \cdot efr, w_7 \cdot cfr, w_8 \cdot pcq, w_9 \cdot ide, w_{10} \cdot rer\}. \tag{17}$$

In addition, the set of mobile application insecurity consequences (MAIC) has the form:

$$MAIC = \{maic_1, \ldots, maic_{15}\} = \\ =\{rd, ml, fr, it, itt, eo, emu, il, pv, ct, ipt, epv, uad, puc, slux\}. \tag{18}$$

Therefore, it is necessary to have the values of the weights of the mobile application insecurity factors. Since the mobile application insecurity consequences are correlated by factors, the existence of such a correlation should be taken into account when calculating the factors' weights.

Since it is necessary to systematize and bring to a single unified form the available information on the mobile application insecurity factors and consequences, as well as to reflect the cause-and-effect relationships between them, let us present a model of the subject area of mobile application security assessment and forecasting in the form of an ontology, taking into account the principles of ontological modelling defined in [38,39].

Formally, an ontology is defined as: O = <X, RX, F>, where X is a finite set of concepts of the subject area, RX is a finite set of relations between concepts, F is a finite set of interpretation functions defined on concepts or relations [38,39].

Then, the model of the subject area of mobile application security assessment and forecasting is as follows: $O_{MAS}$ = <$X_{MAS}$, $RX_{MAS}$, $F_{MAS}$>, where $X_{MAS}$ is a finite set of mobile application insecurity factors and consequences, $RX_{MAS}$ is a finite set of relations between the mobile application insecurity factors and consequences, $F_{MAS}$ is a finite set of interpretation functions defined for the mobile application insecurity factors and consequences. Taking into account the sets (16) and (18), the set of mobile application insecurity factors and consequences has the form:

$$X_{MAS} = \{MAIF, MAIC\} = \{xmas_1, \ldots, xmas_{25}\}, \tag{19}$$

where $\{xmas_1, \ldots, xmas_{10}\}$ included in MAIF, i.e., $\{xmas_1, \ldots, xmas_{10}\}$ = $\{maif_1, \ldots, maif_{10}\}$; $\{xmas_{11}, \ldots, xmas_{25}\}$ included in MAIC, i.e., $\{xmas_{11}, \ldots, xmas_{25}\}$ = $\{maic_1, \ldots, maic_{15}\}$.

The set $RX_{MAS}$ of relations between the concepts consists of the "depend on" relation, i.e., $RX_{MAS}$ = {"depend on"}.

The set $F_{MAS}$ of interpretation functions, defined for the mobile application insecurity factors and consequences, consists of the functions of the dependencies of the mobile application insecurity consequences on the factors, i.e., $F_{MAS}$ = $\{f_1, \ldots, f_{15}\}$.

Then, the ontological model of the subject area of mobile application security assessment and forecasting is as follows:

$$O_{MAS} = <\{xmas_1, \ldots, xmas_{25}\}, \{"depend\ on"\}, \{f_1, \ldots, f_{15}\}>. \tag{20}$$

Since it is necessary to take into account the weights of mobile application insecurity factors when assessing and forecasting the security of mobile applications, we need to introduce such weights into the ontological model of the subject area of assessing and forecasting the security of mobile applications and to develop a weighted ontology.

Considering the sets (17) and (18), the set of mobile application insecurity factors and consequences for the weighted ontology has the form:

$$X_{MASW} = \{MAIFW, MAIC\} = \{xmas_1, \ldots, xmas_{25}\}, \tag{21}$$

where $\{xmasw_1, \ldots, xmasw_{10}\}$ included in MAIFW, i.e., $\{xmasw_1, \ldots, xmasw_{10}\} = \{maifw_1, \ldots, maifw_{10}\}$; $\{xmasw_{11}, \ldots, xmasw_{25}\}$ included in MAIC, i.e., $\{xmasw_{11}, \ldots, xmasw_{25}\} = \{maic_1, \ldots, maic_{15}\}$.

The set $RX_{MASW}$ of relations between concepts also consists only of the "depends on" relation, i.e., $RX_{MASW} = \{$"depend on"$\}$.

The set $F_{MASW}$ of interpretation functions, defined for the mobile application security factors and consequences, consists of the functions of the dependencies of the mobile application insecurity consequences on the weighted factors, i.e., $F_{MASW} = \{\varphi_1, \ldots, \varphi_{15}\}$.

Then, the weighted ontological model of the subject area of mobile application security assessment and forecasting is as follows:

$$O_{MASW} = <\{xmasw_1, \ldots, xmasw_{25}\}, \{\text{"depend on"}\}, \{\varphi_1, \ldots, \varphi_{15}\}>. \tag{22}$$

The developed models of dependencies of mobile application insecurity consequences on the factors (Equations (1)–(15)), as well as the ontological (Equation (20)) and weighted ontological (Equation (22)) models of the subject area, solve the problem of systematizing all the available information on assessing and forecasting the security of mobile applications and bringing it to a single unified form and are necessary to reflect the causal relationships between mobile application insecurity factors and consequences.

### 3.2. Method of Evaluating the Weights of Factors of Mobile Application Insecurity

It is obvious that to develop a weighted ontology of the subject area of mobile application security assessment and forecasting, whose model is represented by Equation (22), it is necessary to have the weights of the factors of mobile application insecurity. Let us develop the method of evaluating the weights of the factors of mobile application insecurity, based on the concept of evaluating the weights of software quality measures and indicators presented by one of the co-authors in [40].

The method of evaluating the weights of factors of mobile application insecurity consists of the following steps:

1. Identifying the common factors for the mobile application insecurity consequences:

    1.1. Formation of a matrix of common factors for the mobile application insecurity consequences $MAICJ = (maicj_{k,l})_{15 \times 15} = \cap_{k=1}^{15} \cap_{l=1}^{15} (MAICM_k, MAICM_l)$, where $maicj_{k,l} = \{MAICM_k \cap MAICM_l\}$ is the k,l-th element of the matrix, which is the set of attributes, which are common to the k-th and l-th mobile application insecurity consequences; $MAICM_k$ and $MAICM_l$ are, respectively, the k-th and l-th mobile application insecurity consequences, represented by the sets of their factors according to the models represented by Equations (1)–(15), but the diagonal elements of the matrix are empty sets, i.e., $maicj_{k,k} = \varnothing$;

    1.2. Formation of a matrix of the number of common factors for the mobile application insecurity consequences $MAICJN = (maicjn_{k,l})_{15 \times 15}$, where $maicjn_{k,l} = |maicj_{k,l}| = |\{MAICM_k \cap MAICM_l\}|$ is the k,l-th element of the matrix, which is equal to the number of elements of the corresponding set $maicj_{k,l}$, i.e., the number of common factors of the k-th and l-th mobile application insecurity consequences;

1.3. Formation of the set of common factors $JF = \{jf_1, \ldots, jf_m\}$ (where m is the number of relevant common factors) for the mobile application insecurity consequences based on the elements of the matrix MAICJ as a symmetric difference (disjunctive sum) of all set elements $maicj_{k,l}$, for which the condition $k < l$ is met (i.e., elements above the main diagonal): $JF = \{maicj_{1,2} \oplus maicj_{1,3} \oplus \ldots \oplus maicj_{k,l} \oplus maicj_{14,15}\}$;

1.4. Formation of the matrix of dependence of the mobile application insecurity consequences from common factors $F = (f_{k,l})_{m \times 15}$, where the k,l-th element of the matrix $f_{k,l} = 1$, if $jf_k \in MAICM_l$, i.e., if the k-th common factor is included in the set of factors of the l-th consequence.

2. Calculation of the weights of the mobile application insecurity factors based on the number of mobile application insecurity consequences that depend on these factors:

2.1. Counting the number of consequences $kc_h$, which depend on the h-th common factor: $kc_h = kc_h + 1$, if $f_{h,l} = 1$ (l = 1 … 15), counting the number of "1s" in each row of the matrix F;

2.2. Calculation of the weight of the h-th factor by the formula: $w_h = kc_h/kf$, where kf is the total number of factors (as shown above, now the mobile application insecurity consequences depend on 10 different factors, i.e., currently, kf = 10); the numerator of the weights of each factor indicates the number of mobile application insecurity consequences that depend on this factor, because if several factors leading to the same consequence are present but not identified or are not accurately determined, the validity of the obtained estimate of such a consequence of mobile application insecurity is significantly reduced, or the possibility of obtaining such an estimate disappears altogether.

A developed method of evaluating the weights of the factors of mobile application insecurity determined the weights of the factors and provides the conclusion about factors, which are necessary to identify and accurately evaluate to ensure an appropriate level of reliability when forecasting and assessing the security of mobile applications.

## 4. Results and Discussion

*4.1. Results: Evaluating the Weights of Factors of Mobile Application Insecurity*

Using the developed method of evaluating the weights of the factors of mobile application insecurity, let us evaluate the weights of ten known factors of mobile application insecurity.

In step 1, we identified the common factors for the mobile application insecurity consequences. We created a MAICJ matrix of common factors for the mobile application insecurity consequences (on the basis of Formulas (1)–(15)).

Next, we formed the matrix MAICJN of the number of common factors for the mobile application insecurity consequences—Table 2.

Let us form the set JF of common factors for the mobile application insecurity consequences; each of the factors was used more than once, so JF = MAIF = {ipu, ids, icsc, iar, iazr, efr, cfr, pcq, ide, rer).

Next, we formed the matrix F of the dependence of the mobile application insecurity consequences on common factors—Table 3.

In step 2, we calculated the weights of the mobile application insecurity factors based on the number of mobile application insecurity consequences that depended on these factors. Let us start by counting the number of consequences that depend on each common factor: $kc_1 = 4$ for factor ipu; $kc_2 = 5$ for factor ids; $kc_3 = 3$ for factor icsc; $kc_4 = 3$ for factor iar; $kc_5 = 3$ for factor iazr; $kc_6 = 4$ for factor efr; $kc_7 = 7$ for factor cfr; $kc_8 = 5$ for factor pcq; $kc_9 = 6$ for factor ide; $kc_{10} = 6$ for factor rer.

Then, we calculated the weight of each factor:
$w_1 = kc_1/kf = 4/10 = 0.4$ for factor ipu;
$w_2 = kc_2/kf = 5/10 = 0.5$ for factor ids;
$w_3 = kc_3/kf = 3/10 = 0.3$ for factor icsc;

$w_4 = kc_4/kf = 3/10 = 0.3$ for factor iar;
$w_5 = kc_5/kf = 3/10 = 0.3$ for factor iazr;
$w_6 = kc_6/kf = 4/10 = 0.4$ for factor efr;
$w_7 = kc_7/kf = 7/10 = 0.7$ for factor cfr;
$w_8 = kc_8/kf = 5/10 = 0.5$ for factor pcq;
$w_9 = kc_9/kf = 6/10 = 0.6$ for factor ide;
$w_{10} = kc_{10}/kf = 6/10 = 0.6$ for factor rer.

The numerator of the weights of each factor indicated the number of mobile application insecurity consequences that depended on that factor.

**Table 2.** Matrix MAICJN of the number of common factors for the mobile applications insecurity consequences.

|      | rd | ml | fr | it | itt | eo | emu | il | pv | ct | ipt | epv | uad | puc | slux |
|------|----|----|----|----|-----|----|-----|----|----|----|-----|-----|-----|-----|------|
| rd   | 0  | 3  | 2  | 1  | 5   | 2  | 1   | 2  | 1  | 2  | 2   | 2   | 3   | 3   | 1    |
| ml   | 3  | 0  | 2  | 2  | 1   | 1  | 0   | 1  | 1  | 1  | 1   | 2   | 2   | 1   | 0    |
| fr   | 2  | 2  | 0  | 2  | 1   | 0  | 0   | 0  | 1  | 0  | 0   | 1   | 0   | 0   | 0    |
| it   | 1  | 2  | 2  | 0  | 0   | 0  | 0   | 0  | 1  | 0  | 0   | 1   | 0   | 0   | 0    |
| itt  | 5  | 1  | 1  | 0  | 0   | 1  | 0   | 1  | 1  | 2  | 1   | 0   | 2   | 2   | 0    |
| eo   | 2  | 1  | 0  | 0  | 1   | 0  | 1   | 0  | 0  | 1  | 0   | 0   | 1   | 2   | 1    |
| emu  | 1  | 0  | 0  | 0  | 0   | 1  | 0   | 0  | 0  | 0  | 0   | 0   | 0   | 1   | 1    |
| il   | 2  | 1  | 0  | 0  | 1   | 0  | 0   | 0  | 1  | 1  | 2   | 1   | 1   | 0   | 0    |
| pv   | 1  | 1  | 1  | 1  | 1   | 0  | 0   | 1  | 0  | 1  | 2   | 1   | 1   | 0   | 0    |
| ct   | 2  | 1  | 0  | 0  | 2   | 1  | 0   | 1  | 1  | 0  | 1   | 0   | 1   | 1   | 0    |
| ipt  | 2  | 1  | 0  | 0  | 1   | 0  | 0   | 2  | 2  | 1  | 0   | 2   | 2   | 0   | 0    |
| epv  | 2  | 2  | 1  | 1  | 0   | 0  | 0   | 1  | 1  | 0  | 2   | 0   | 2   | 0   | 0    |
| uad  | 3  | 2  | 0  | 0  | 2   | 1  | 0   | 1  | 1  | 1  | 2   | 2   | 0   | 1   | 0    |
| puc  | 3  | 1  | 0  | 0  | 2   | 2  | 1   | 0  | 0  | 1  | 0   | 0   | 1   | 0   | 1    |
| slux | 1  | 0  | 0  | 0  | 0   | 1  | 1   | 0  | 0  | 0  | 0   | 0   | 0   | 1   | 0    |

**Table 3.** Matrix F of the dependence of the mobile application insecurity consequences on common factors.

|      | rd | ml | fr | it | itt | eo | emu | il | pv | ct | ipt | epv | uad | puc | slux |
|------|----|----|----|----|-----|----|-----|----|----|----|-----|-----|-----|-----|------|
| ipu  |    | 1  | 1  | 1  |     |    |     |    | 1  |    |     |     |     |     |      |
| ids  | 1  | 1  | 1  | 1  |     |    |     |    |    |    |     |     | 1   |     |      |
| icsc | 1  |    |    |    | 1   |    |     |    |    |    |     |     | 1   |     |      |
| iar  | 1  |    |    |    | 1   |    |     |    |    |    |     |     |     |     | 1    |
| iazr | 1  |    | 1  |    | 1   |    |     |    |    |    |     |     |     |     |      |
| efr  |    |    |    |    |     |    |     |    | 1  |    |     | 1   | 1   | 1   |      |
| cfr  | 1  | 1  |    |    | 1   | 1  |     |    |    |    | 1   |     |     | 1   | 1    |
| pcq  | 1  |    |    |    |     | 1  | 1   |    |    |    |     |     |     | 1   | 1    |
| ide  | 1  |    |    |    | 1   |    |     | 1  | 1  | 1  | 1   |     |     |     |      |
| rer  | 1  | 1  |    |    |     |    |     | 1  |    |    | 1   | 1   | 1   |     |      |

Taking into account the determined weights of each factor, the set of weighted factors of the mobile applications insecurity (Equation (17)) had the following form:

$$\text{MAIFW} = \{maifw_1, \ldots, maifw_{10}\} = \{0.4 \cdot ipu, 0.5 \cdot ids, 0.3 \cdot icsc, 0.3 \cdot iar, 0.3 \cdot iazr, \\ 0.4 \cdot efr, 0.7 \cdot cfr, 0.5 \cdot pcq, 0.6 \cdot ide, 0.6 \cdot rer). \tag{23}$$

In this paper, the weights of the factors of mobile application insecurity were evaluated, taking into account the correlations of the mobile applications insecurity consequences with the factors, and the weights for ten known factors were determined. However, such an assessment of the weights of the factors of mobile application insecurity is easily scalable and adaptable to changes in the number and list of factors and consequences of mobile application insecurity.

### 4.2. Results: Identifying the Mutual Correlations of Mobile Application Insecurity Factors and Consequences

Considering that one factor can cause several of the consequences obtained in Section 4.1's results, we decided to conduct an analysis to find out which factors caused the same consequences and which factors caused the most consequences in order to identify the most potentially dangerous factors. The diagram of consequences of mobile application insecurity dependency on the factors that affected mobile application security is presented in Figure 2. From Figure 2 we can see that all factors can cause three or more consequences. Thus, improper platform usage caused four consequences. Insecure data storage caused five consequences. Insecure client–server communication caused three consequences. Insecure authentication risk caused three consequences. Insecure authorization risk caused three consequences. Extraneous functionality risk caused four consequences. Code forgery risk caused seven consequences. Poor code quality caused five consequences. Insufficient data encryption and reverse engineering risk caused six consequences each. Therefore, we can identify that three factors that cause the most consequences can be considered as the most potentially dangerous for mobile application security—code forgery risk, insufficient data encryption, and reverse engineering risk.

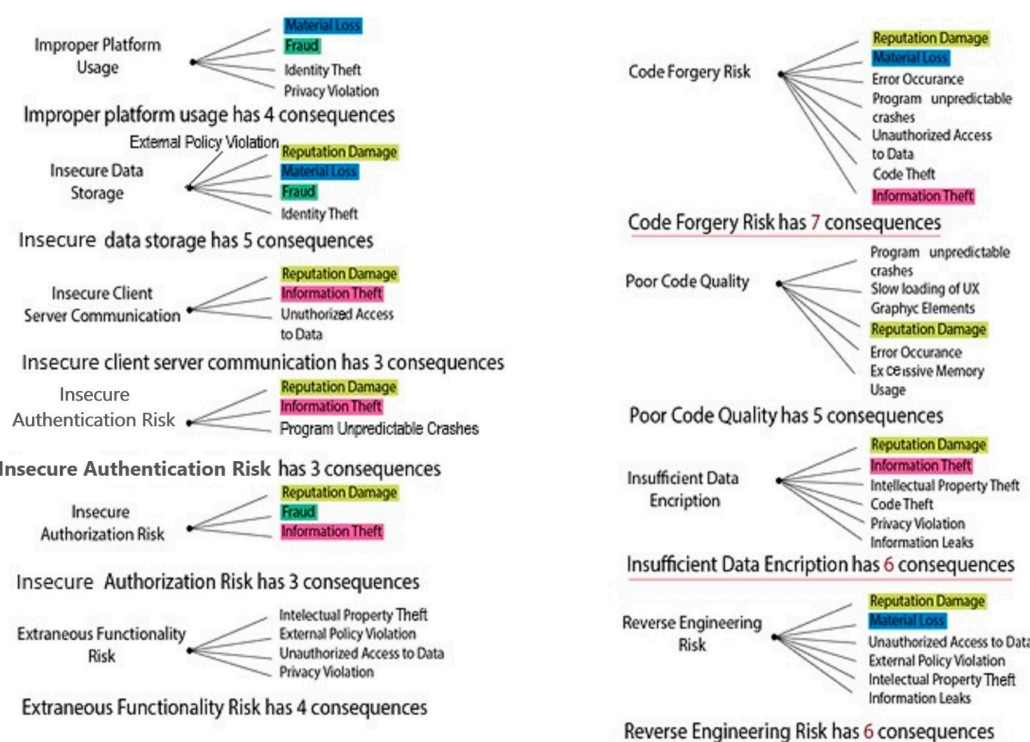

**Figure 2.** Diagram of Consequences of Mobile Application Insecurity Dependency on Factors That Affect Mobile Application Security.

To see the correlation between the factors and the consequences that they cause, we created a diagram of the correlations between factors and consequences, which is presented in Figure 3. The diagram of the correlations between factors and consequences shows two types of relationships between the mobile application insecurity factors and consequences—the "cause" relationship, which reflects which factors cause which consequences, and the "depend on" relationship, which reflects which consequences depend on which factors.

From the correlation diagram, we can see that not only one factor can cause several consequences, but also that one consequence can depend on more than one factor. Let us analyze the diagram in Figure 3 and determine which consequences depend on more than one factor.

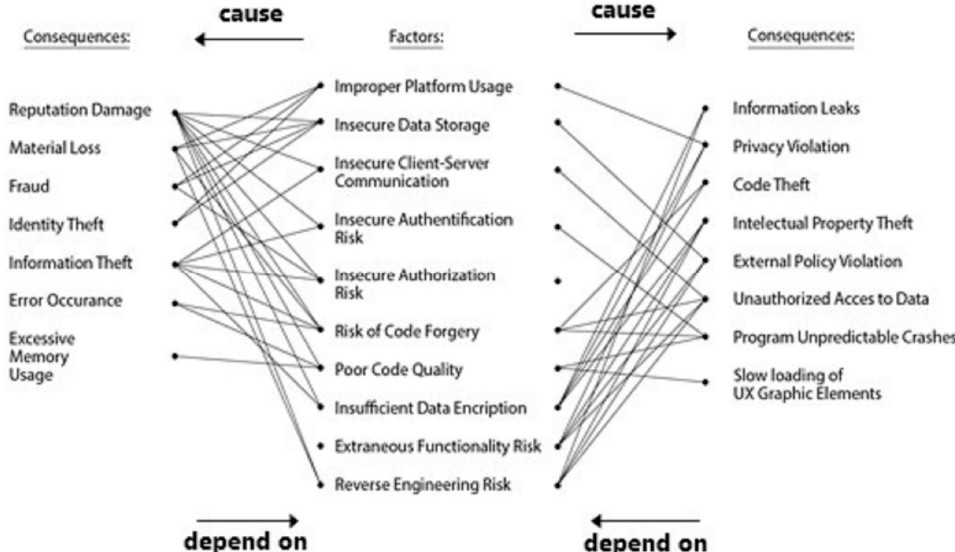

**Figure 3.** Diagram of the Correlations Between Factors and Consequences.

Thus, material loss is caused by four factors. Fraud is caused by three factors. Reputation damage is caused by eight factors. Identity theft is caused by two factors. Information theft is caused by five factors. Error occurrence is caused by two factors. Excessive memory usage is caused by one factor. Information leaks are caused by two factors. Privacy violation is caused by three factors. Code theft is caused by two factors. Intellectual property theft is caused by three factors. External policy violation is caused by two factors. Unauthorized access to data is caused by five factors. Program unpredictable crashes are caused by one factor and a slow loading of UX graphic elements is also caused by one factor.

Special attention should be paid to consequences affected by more than one threat factor. They are the ones that pose the greatest danger to users, for example, unauthorized access to data, which can be caused by four threat factors (Figure 4a). This is dangerous both for users, as their personal data may end up in the hands of attackers, and for the company, with the risk of lawsuits from users and, as a result, significant financial losses. The risk of reputation damage can be caused by eight threat factors (Figure 4b). This is dangerous for the company's founder and the developer of the mobile application, as this consequence is irreversible. The company loses not only the trust of users but also incurs significant financial losses.

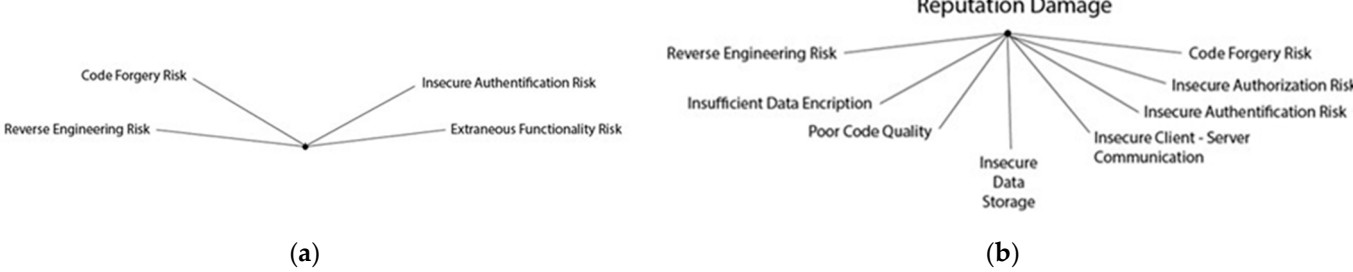

(**a**)　　　　　　　　　　　　　　　　　　　　　(**b**)

**Figure 4.** Consequences Caused by the Largest Number of Threat Factors: (**a**) Unauthorized Access to Data is Caused by 4 Threat Factors; (**b**) Reputation Damage is Caused by 8 Threat Factors.

Thus, this subsection identified the mutual correlations of mobile application insecurity consequences with factors, identified common factors on which more than one consequence depended in order to increase the accuracy of their values when assessing and forecasting the mobile application security.

*4.3. Discussion*

As it was proved above, when developing effective methods and tools for assessing and forecasting the security of mobile applications, there is a constant need to forecast the consequences to assess, for example, the probability of their occurrence based on assessments of existing factors. The existence of relationships (correlations) between the mobile application insecurity consequences and the factors affects the significance and weights of the factors.

If one or several factors that lead to the same consequence are present but not identified or are not accurately determined, the validity of the obtained estimate of such a consequence of mobile application insecurity is significantly reduced or the possibility of obtaining such an estimate disappears altogether. Therefore, it is important to mitigate the impact of the mutual correlations of such mobile application insecurity consequences with factors, through ensuring and increasing the accuracy of their values; for this, namely, the identification of common factors, counting the number of mobile application insecurity consequences that depend on these factors is necessary. The conducted study was devoted to identifying the mutual correlations and evaluating the weights of the factors and consequences of mobile application insecurity.

Since there is a need to systematize and unify the available information on mobile application insecurity factors and consequences, as well as to reflect the cause-and-effect relationships between them (for example, the relationship (correlation) between the consequences of mobile application insecurity and the factors), it was decided to use the apparatus of ontologies and weighted ontologies, which allowed us to specify the information on mobile application insecurity factors and consequences necessary to describe and solve the problems of this subject area. In this paper, the ontologies and weighted ontologies provided a theoretical basis for the development of methods and tools for assessing, forecasting, and ensuring the security of mobile applications.

The advantage of using ontologies and weighted ontologies for organizing the information about mobile application insecurity factors and consequences is the possibility of an automatic processing of this information by ontology-based intelligent agents, as well as automatically assessing and forecasting the mobile application's security based on the received information.

## 5. Conclusions

Currently, there is a contradiction between the growing number of mobile applications in use and the responsibility that is placed on them, on the one hand, and the imperfection of the methods and tools for ensuring the security of mobile applications, on the other hand. Thus, ensuring the security of mobile applications by developing effective methods and tools is a challenging task today. The purpose of this research was to evaluate the mutual correlations and weights of the factors and consequences of mobile application insecurity.

We developed models of dependencies of mobile application insecurity consequences on the factors, as well as ontological and weighted ontological models of the subject area, which solved the problem of systematizing all the available information for assessing and forecasting the security of mobile applications and bringing it to a single unified form and were necessary to reflect the causal relationships between mobile application insecurity factors and consequences.

A method of evaluating the weights of factors of mobile application insecurity was developed, which, taking into account the mutual correlations of mobile application insecurity consequences with the factors, determined the weights of the factors and allowed us to conclude which factors were necessary to identify and accurately determine (evaluate) to ensure an appropriate level of reliability for forecasting and assessing the security of mobile applications. The developed method for evaluating the weights of the factors of mobile application insecurity is easily scalable and adaptable to changes in the number and list of factors and consequences of mobile application insecurity.

The paper evaluated the weights of factors of mobile application insecurity and determined the weights for ten factors known to date. The paper identified the mutual correlations of the consequences of mobile application insecurity with these factors, identified the common factors on which more than one consequence depended in order to increase the accuracy of their values when assessing and predicting the security of mobile applications.

The representativeness of a research study and its results is achieved through a proper design of the study, which should reproduce the general object of research in terms of the parameters essential to the study. In our case, the object of research was the OWASP mobile application insecurity factors and mobile application insecurity consequences that depend on these factors. The study was devoted to evaluating and weighting these selected ten factors and the mutual correlations of the consequences of mobile application insecurity with these factors. The experimental results of our research are the evaluation of the weights of ten OWASP mobile application insecurity factors, the identification of the mutual correlations of the consequences of mobile application insecurity with these factors, and the identification of common factors on which more than one consequence depends. Thus, this study, as well as its results, is representative, as it correctly reproduces the general object of our study.

The areas for future research by the authors are:

1. The implementation of the ontology and weighted ontology of the subject area of assessing and forecasting the security of mobile applications, represented by Equations (20) and (22), respectively, using, for example, the Protégé platform;
2. Establishing the dependencies of the mobile application insecurity consequences on the factors—the form of functions $f_1$–$f_{15}$, $\varphi_1$–$\varphi_{15}$, which are currently unknown;
3. The design and implementation of ontology-based intelligent agents that will provide the ability to automatically process information on the subject area of assessing and forecasting the security of mobile applications, as well as the ability to automatically assess and forecast the security of mobile applications based on the received information;
4. The design and development of methods and tools for forecasting, assessing, and ensuring the security of mobile applications;
5. The research of other (in addition to OWASP) factors that affect mobile application security, the search for their mutual correlations, the calculation of their weights, and adding them to the developed ontologies.

**Author Contributions:** Conceptualization, E.Z. and T.H.; methodology: O.P. and Y.V.; state of the art, T.H. and O.P.; results, O.P. and Y.V.; discussion, E.Z. and T.H.; writing—original draft preparation, T.H. and O.P.; writing—review and editing, E.Z.; visualization, O.P.; supervision, E.Z.; project administration, T.H.; funding acquisition, T.H. All authors have read and agreed to the published version of the manuscript.

**Funding:** This research received no external funding.

**Data Availability Statement:** The dataset used for the findings is included in the manuscript.

**Acknowledgments:** The investigation in this paper was partly implemented under the project "New methods development for reliability analysis of complex system" (APVV-18-0027).

**Conflicts of Interest:** The authors declare no conflict of interest.

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
