# Peer review of "Identifying the Mutual Correlations and Evaluating the Weights of Factors and Consequences of Mobile Application Insecurity"

_systems, doi:10.3390/systems11050242_

Round 1

Reviewer 1 Report

In this paper, the author evaluates the weights of 10 mobile application insecurity factors and identifies the mutual correlations of the consequences of mobile applications insecurity by factors. However, you should make a modification according to the following suggestions to publish this paper.

1.     The structure of the abstract is not reasonable. The methodology proposed in this paper is heavily elaborated, however, the research background and experimental results are lacking, please revise it.

2.     In the Introduction section, previous scholars' research about mobile application risk is ignored.

3.     Section 2 is unreasonable. Previous scholars' research and modeling on the unsafe factors of mobile applications should be included in section 2. However, in the paper, only 10 the most prominent factors that affect mobile applications security are analyzed.

4.     There is a lot of blank area under table 1, similarly, there is a lot of blank under formula 16, please delete it.

5.     Figure 2 is unclear, please redraw it.

6.     In Equation 1, the meanings of different characters are not properly expressed. For example, rd – reputation damage, which is Chinese style expression, you should use a sentence to express the meaning of rd. Other similar expressions in the paper should also be modified.

7.     In Equation 16, the meaning of MAIF is ignored. In Equation 17, the meaning of MAIFW is ignored. In Equation 18, the meaning of MAIC is ignored.

8.     In section 3.2, the weight that influence the mobile applications insecurity consequences is calculated by the formula: wh = kch / kf. Please give reasons why you define weights in this way.

9.     The size of Tables 2-4 is too large compared to the content of the paper, which is unattractive, please redraw it.

10.  In section 4.3, the discussion is too complicated, you should revise it to highlight the key points of this discussion.

11.  The references are not up to date. You should add the following references.

i)      https://doi.org/10.1049/ise2.12106

ii)     https://doi.org/10.1049/ise2.12105

iii)    https://doi.org/10.1109/TEM.2021.3066090

iv)   https://doi.org/10.1016/j.compedu.2022.104449

v)     http://doi.org/10.1109/ACCESS.2022.3159679

vi)   https://doi.org/10.1016/j.jpdc.2022.01.002

Author Response

Dear Reviewer! Great thanks for your comments! Point-by-point response to your comments is in the attached file

Reviewer 2 Report

The authors identify mutual correlations and evaluate the weights of factors and consequences of mobile applications insecurity based on ten examples. While I like the approach, I'm unsure how the examples were selected and if they are representative. Hence, this also follows for the results. You should describe your methodology and explain, why they are representative.

In the introduction, you use sources from 2018 for statistics. Since then, some things might have changed. You might search for newer sources and compare them with the statistics from 2018. This might provide additionally provide trends. Similarly, you could provide more current examples in Table 1.

I'm unsure how you derived the numbers from Figure 1.

In Section 2, it stays unclear if other publications have looked into app insecurity.

Minor comments:
- Strengthen your abstract
- Describe your contribution in the introduction.
- First paragraph in the Introduction: What about Apple's procedure before apps are released in the App Store?

Table 1: WhatsApp Messenger with one "t"

You typically should not use any shortened words such as don't. Use do not instead.

Figures: please make sure that you don't have any spelling mistakes in the embedded text.

You typically write out numbers up to (including) twelve.

subsection 4.1 -> Section 4.1. If you reference a specific element, then the name of the element starts with a big letter. You don't have anything below a section.

In Section 4.1, you often use "let's".

Author Response

(The authors gave the same response as above.)

Round 2

Reviewer 2 Report

Thank you for incorporating the comments.

Minor comment: it is better style to name the authors (X and Y [n]) instead of saying "the paper [n]).